# Arithmetic Optimization AOMDV Routing Protocol for FANETs

**DOI:** 10.3390/s23177550

**Published:** 2023-08-31

**Authors:** Huamin Wang, Yongfu Li, Yubing Zhang, Tiancong Huang, Yang Jiang

**Affiliations:** 1School of Microelectronics and Communication Engineering, Chongqing University, Chongqing 401331, China; 202112131155@cqu.edu.cn (H.W.); htc@cqu.edu.cn (T.H.); 2State Grid Chongqing Electric Power Company, Chongqing 401123, China; liyongfu1@cq.sgcc.com.cn; 3Beijing Smart-Chip Microelectronics Technology Co., Ltd., Beijing 100005, China; zhangyubing@sgchip.sgcc.com.cn

**Keywords:** FANETs, arithmetic optimization, AOMDV, AO-AOMDV

## Abstract

Flying ad hoc networks (FANETs), composed of small unmanned aerial vehicles (UAVs), possess characteristics of flexibility, cost-effectiveness, and rapid deployment, rendering them highly attractive for a wide range of civilian and military applications. FANETs are special mobile ad hoc networks (MANETs), FANETs have the characteristics of faster network topology changes and limited energy. Existing reactive routing protocols are unsuitable for the highly dynamic and limited energy of FANETs. For the lithium battery-powered UAV, flight endurance lasts from half an hour to two hours. The fast-moving UAV not only affects the packet delivery rate, average throughput, and end-to-end delay but also shortens the flight endurance. Therefore, research is urgently needed into a high-performance routing protocol with high energy efficiency. In this paper, we propose a novel routing protocol called AO-AOMDV, which utilizes arithmetic optimization (AO) to enhance the ad hoc on-demand multi-path distance vector (AOMDV) routing protocol. The AO-AOMDV utilizes a fitness function to calculate the fitness value of multiple paths and employs arithmetic optimization for selecting the optimal route for routing selection. Our experiments were conducted using NS3 with three evaluation metrics: the packet delivery ratio, network lifetime, and average end-to-end delay. We compare this algorithm to routing protocols including AOMDV and AODV. The results indicate that the proposed AO-AOMDV attained a higher packet delivery ratio, network lifetime, and lower average end-to-end delay.

## 1. Introduction

With developments in science and technology, unmanned aerial vehicles (UAVs) are more widely used in military and civil fields, such as inspections of Safety Critical Infrastructure [1], precision agriculture [2], rescuing during disaster [3], intelligent logistics [4], urban traffic patrol [5], and so on. No matter in which application scenario, due to the limitations of its own energy and payload, it is challenging for a single unmanned aerial vehicle to accomplish complex tasks. Therefore, in order to overcome the shortcomings of a single UAV system, the collaboration and establishment of mobile ad hoc networks (MANETs) among multiple UAV nodes are referred to as flying ad hoc networks (FANETs).

The FANETs, as a special class of MANET, directly use the traditional MANET routing protocol. Similar to MANETs, depending on whether the geographic location information is needed, the routing protocols of the FANETs can be divided into topology-based routing protocols [6] and location-based routing protocols [7].

We generally categorize topology-based routing protocols into reactive routing, proactive routing, and hybrid routing protocols. Reactive routing involves initiating route requests only when the source node needs to communicate with a destination node. Common reactive routing protocols in recent years include dynamic source routing (DSR) [8,9,10,11] and the ad hoc on-demand distance vector routing protocol (AODV) [12,13,14,15,16]. Proactive routing protocols, on the other hand, detect changes in network topology by regularly broadcasting routing update messages to other nodes in order to update their routing tables. Well-known proactive routing protocols in recent years include the destination-sequenced distance vector (DSDV) [12,17], optimized link state routing (OLSR) [18,19], and the wireless routing protocol (WRP) [20,21]. Finally, hybrid routing protocols aim to leverage the respective advantages of reactive routing and proactive routing. The zone routing protocol (ZRP) [22,23,24] and zone-based hierarchical link state routing (ZHLS) [25,26] prescribe the use of proactive routing within a designated zone for each node, while employing an on-demand routing mechanism similar to DSR for routing to nodes outside the zone. However, the performance of hybrid routing is largely determined by the value of the zone radius parameter. Therefore, its adaptability is limited.

Unlike topology-based routing protocols, location-based routing utilizes the location information of current and neighboring nodes for forwarding decisions as the greedy perimeter stateless routing (GPSR) [27], geographical- and energy-aware routing (GEAR) [28], and graph embedding (GEM) [29]. The premise of location-based routing protocols is that UAV nodes are able to obtain their own location information through GPS. In location-based routing, nodes do not perform global topology discovery but rely on local information for communication. However, location-based routing introduces challenges such as route failures and routing loops, requiring the design of corresponding strategies to address these issues.

Meta-heuristics is an optimization method that utilizes the natural phenomena of elements to gradually discover the optimal solution from a set of initial solutions. In the ever-changing and complex environment, meta-heuristic algorithms are more efficient and stable. In MANETs, the application of meta-heuristic optimization algorithms in routing protocols has been widely studied. Due to the characteristics of UAVs, such as fast node mobility, limited energy, dynamic topology, and complex environment, communication between UAV nodes needs to have high reliability and energy efficiency so that the flying formation can make an appropriate flight control strategy for flight formation. Therefore, many studies have proposed meta-heuristic-based routing protocols that use the ant colony [30], particle swarm [31,32], whale [33], and genetic [34] to find the optimal path with higher energy efficiency.

Most of the meta-heuristic algorithms proposed as routing protocols for FANETs have been proposed as a hybrid with the known multi-path routing protocols as the ad hoc on-demand multi-path distance vector (AOMDV) and MP-DSR. Their objective is to achieve a balance between the shortest path, load balancing, and energy conservation of nodes in order to reduce the number of dead nodes and the network overhead, thereby obtaining the optimal route. Arithmetic optimization (AO) [35] has been one of the most investigated meta-heuristic optimization algorithms in the last two years in different fields, including solving mechanical engineering design problems [36], damage assessment in composite plates [37], robot path planning [38], and image thresholding [39]. As far as we know AO has not been investigated before as a routing protocol in FANETs. In this study, we propose an energy-efficient hybrid routing protocol called the arithmetic optimization (AO)-based AOMDV (AO-AOMDV) by utilizing the powerful meta-heuristic algorithm AO. The protocol has demonstrated highly promising results in various research studies, indicating its significant potential.

Although there are variations of meta-heuristic algorithms in the field of population-based optimization methods, their optimization processes generally consist of two main stages: exploration and exploitation. The former refers to the extensive use of algorithmic search agents to cover the search space, thereby avoiding local solutions. The latter improves the accuracy of the solutions obtained during the exploration phase.

In this paper, we introduce a new fitness function on the basis of the AOMDV routing protocol to replace the minimum hop count decision mechanism. Additionally, we propose the utilization of arithmetic optimization to rapidly find the optimal path from paths with different fitness values. We refer to the routing protocol described above as AO-AOMDV. AO-AOMDV incorporates not only the fitness function but also employs arithmetic optimization. The fitness function is utilized to provide the decision mechanism, while arithmetic optimization is dedicated to swiftly implementing the decision mechanism. Within AO-AOMDV, we select the most suitable route from the source to the destination based on three parameters. The chosen route should exhibit relatively high link stability and residual energy while having comparatively lower traffic to avoid congestion.

The structure of the paper is organized as follows: Section 2 covers a survey on the literature review. Section 3 presents the routing issues encountered in the current phase of UAV development. It elaborates on the fitness function and explains the solution by describing the routing discovery and maintenance process. Section 4 outlines the performance evaluation and presents the experimental results. Finally, Section 5 concludes the research outcomes.

## 2. Related Works

As mentioned in the previous section, meta-heuristics play an important role in the research field of FANETs’ routing protocols and energy optimization. Therefore, this section aims to present some recent studies that discuss the application of meta-heuristics in FANETs. Ref. [40] proposed an ad hoc on-demand multi-path distance vector routing protocol-based elephant-herding optimization to maximize the network lifetime. Ref. [41] constructed an ad hoc on-demand multi-path distance vector routing protocol using an algorithm for selecting a consistent path to improve the performance of quality of service (QoS) in high-speed MANETs.

The FF-AOMDV, which introduces an approach for selecting an efficient path that achieves both the shortest distance and the lowest energy consumption, was proposed in [42]. Similar to the AOMDV, in the event of a link disruption, the source node in the FF-AOMDV will utilize the next shortest path to the destination node from the routing table. However, compared to the AOMDV, although the FF-AOMDV considers both energy and shortest distance criteria, its transmission performance is not high, and the improvement in the network lifetime is not significant.

Ref. [43] proposed the QoS-AOMDV protocol, which is based on the AOMDV and aims to enhance QoS support. The QoS-AOMDV obtains high-quality paths by acquiring information on the queue length and remaining energy through cross-layer interactions. However, data collisions result in increased end-to-end delays and decreased packet delivery rates. Lin et al. [44] proposed a multi-path routing protocol that ensures link stability. The protocol defines the link stability probability based on both the mobility model and queue length. This approach effectively balances the distance between nodes and the routing lifetime, thereby enhancing link stability. Nevertheless, due to frequent path changes, a larger number of nodes are involved in maintaining network stability, resulting in higher energy consumption for the nodes.

In [45], a study was conducted on the problem of selecting the optimal path using a hybrid optimization approach based on the 2-Opt algorithm and the Artificial Bee Colony (ABC) algorithm. The 2-Opt algorithm is an optimization method commonly used for solving the Traveling Salesman Problem, while the ABC algorithm simulates the foraging behavior of honeybees to find optimized solutions. To address the path selection problem, fuzzy rules were employed to classify nodes based on their end-to-end delay, thus preventing potential data packet losses that could occur when nodes are on the verge of leaving the network. The simulation results indicate that the proposed fuzzy rule selection in combination with the ABC–2Opt algorithm offers limited improvements in terms of the packet delivery rate and malicious node detection rate compared to the rule selection using the ABC algorithm.

A protocol called the expected remaining lifetime-based AOMDV (ERL-AOMDV) is proposed in [46]. In the latest developments of the AOMDV (ad hoc on-demand multipath distance vector) protocol, the MMRE-AOMDV (minimum–maximum residual energy AOMDV) stands out as the most advanced approach from an energy efficiency perspective. The ERL-AOMDV (energy and residual lifetime AOMDV) goes even further in-depth in studying a node’s residual energy. It emphasizes selecting the optimal path by continuously considering the remaining energy of nodes, estimating the time needed for completing communication sessions, and taking into account the expected remaining lifetime of nodes. The ERL-AOMDV utilizes three optimal paths to send data packets. Although the ERL-AOMDV showed significant improvements in performance across different node densities, it did not take into account performance under varying UAV velocities.

## 3. AO-AOMDV Routing Protocol

### 3.1. Problem Statement

With the rapid development of the UAV industry, there are fewer routing protocols available for high-speed mobile UAV swarms. The majority of the proposed research studies have focused on the performance and lifetime of FANETs with respect to changes in density while neglecting the significant impact of velocity changes on UAV swarms. Therefore, it is necessary to consider how to enhance the routing performance and network lifetime of UAV swarms at varying velocities. We not only need to select stable and optimal paths but also ensure a minimal routing processing time in order to increase the packet delivery rate and network lifetime and reduce the end-to-end delay. Thus, it is necessary to consider a routing protocol that utilizes optimization algorithms for route selection during the communication process.

### 3.2. Proposed Method

The source nodes use the AOMDV protocol search for multiple routes to the destination node by sending route request (RREQ) packets without considering other factors of the routes and only selecting the route with the minimum number of hops as the transmission path. In this regard, this paper introduces a new fitness function based on the AOMDV protocol and utilizes arithmetic optimization for route selection. In the proposed method, when multiple routes to the destination node are found, the source node needs to consider the possibility of packet loss caused by link failures and thus selects a stable and optimized route with higher residual energy and lower traffic. In other words, the fitness function will consider the following factors:The congestion situation of each node in the routing;The residual energy of each node in the route;The link holding time for each path in the routing.

The selection of the route from the source node to the destination node is based on the highest fitness value of the route. The main criteria followed by the optimal route are: (a) having the highest residual energy, (b) having the highest link holding time, and (c) containing lower traffic. The source node sends data packets through a stable route with the highest residual energy and fewer congested nodes.

In the proposed system, the UAV nodes are uniformly distributed in three-dimensional space to form a FANET. The following assumptions are made for the FANETs:The mobility of nodes may lead to link disconnections and time delays;All moving nodes in the FANETs are initialized with equal energy and quality;Nodes exhibit random mobility, resulting in constantly changing distances between nodes;FANETs consist of mobile nodes, each having a unique identification number.

### 3.3. Fitness Function

This paper proposes a solution that optimizes routes using an arithmetic optimization algorithm. In the beginning, this paper introduces a new fitness function based on three components as follows. The first component considers the link holding time between nodes. The second component considers the residual energy of the nodes, and the third component considers the congestion degree of the nodes.

Link Holding Time:Assuming that the drone node can obtain its current location and speed information by receiving GPS signals, the communication duration between two nodes can be predicted based on the current location and speed information of the nodes. The model for predicting the keep-alive time is as follows:
(1)$$\begin{array}{c} \begin{array}{cc} & HT_{MN}=\frac{\sqrt{\alpha R^{2}-\beta }}{\alpha } \end{array} \end{array}$$
where *R* represents the communication range of a node. The coordinates of node M are $\left(x_{1},y_{1},z_{1}\right)$, and its velocities along the axes are $\left(v_{1x},v_{1y},v_{1z}\right)$. The coordinates of node N are $\left(x_{2},y_{2},z_{2}\right)$, and its velocities along the axes are $\left(v_{2x},v_{2y},v_{2z}\right)$. Let $a=x_{2}-x_{1},b=v_{2x}-v_{1x},c=y_{2}-y_{1},d=v_{2y}-v_{1y},e=z_{2}-z_{1},f=v_{2z}-v_{1z}$, $\alpha =b^{2}+d^{2}+f^{2},\beta ={\left(ad-bc\right)}^{2}+{\left(af-be\right)}^{2}+{\left(ed-cf\right)}^{2}.$

Residual Energy:When the MANETs’ nodes communicate wirelessly with each other, there are many factors that affect their energy consumption rate. Among these, the most critical factor is the operating mode of the WiFi device. Under different operating modes, the wireless channel between nodes has different physical layer states, which correspond to different antenna reception and transmission powers.Taking the IEEE 802.11 physical layer and MAC layer interaction protocol, which is the most commonly used in networking, as an example, the energy consumption of a node’s wireless communication is mainly determined by the six states defined by the 802.11 protocol. Different device states correspond to different working currents. Therefore, the energy consumed by a wireless network card within a certain working time can be expressed as Equation (Equation 2).
(2)$$\begin{array}{c} \begin{array}{ccc} E_{0} & = & \sum_{i=1}^{6}\left(P_{{\mathrm{Status}}_{i}}\times T_{{\mathrm{Status}}_{i}}\right)\\ & = & U\times \sum_{i=1}^{6}\left(I_{{\mathrm{Status}}_{i}}\times T_{{\mathrm{Status}}_{i}}\right) \end{array} \end{array}$$
when i=1∽6, $Status_{i}$ can be one of the following states: IDLE, CCA BUSY (clear channel assessment busy), Tx, Rx, Switching, or Sleep. $P_{{\mathrm{Status}}_{i}}$ and $I_{{\mathrm{Status}}_{i}}$ represent the power and current consumption corresponding to each Statusi state, respectively. *U* is the rated voltage for the wireless network card. $T_{{\mathrm{Status}}_{i}}$ is the total time that the wireless network card stays in the $Status_{i}$ state during the time interval $T_{0}$.
(3)$$\begin{array}{c} \begin{array}{cc} & T_{0}=\sum_{i=1}^{6}T_{Status_{i}} \end{array} \end{array}$$The node calculates its remaining energy at time *t* using the following equation:
(4)$$\begin{array}{c} \begin{array}{cc} & E_{current}=E_{initial}-E_{0} \end{array} \end{array}$$
where $E_{initial}$ represents the initial energy of the node.

Congestion Degree:Congestion can cause increased network delays, packet loss, and energy consumption. To address this problem, this paper proposes a congestion detection method: using the ratio of the number of packets cached in the MAC layer interface queue to the maximum length of the interface queue as a measure of the current node load. The formula for calculating the current payload congestion degree, denoted as $C_{payload}$, of node *j* on path *i* is as follows:
(5)$$\begin{array}{c} \begin{array}{cc} & C_{payload}=Q_{c}/Q_{l} \end{array} \end{array}$$
where $Q_{c}$ represents the number of packets cached in the MAC layer interface queue of node *j*, and $Q_{l}$ represents the maximum number of packets that the MAC layer interface queue of node *j* can accommodate the length of the interface queue. The length of the interface queue is the maximum packet length set for each node’s buffer, and its value is fixed. As packets enter or leave the network, the number of packets cached in the MAC layer interface queue of a node changes continuously, so the ratio of the number of packets cached in the MAC layer interface queue to the length of the interface queue varies at different times, indicating different congestion degrees of the node.

The fitness function in the AO-AOMDV considers the link holding time, residual energy, and congestion degree, and the fitness function of path $p_{i}$ can be represented as:(6)$$\begin{array}{c} \begin{array}{cc} & F\left(p_{i}\right)=\omega_{2}HT\left(p_{i}\right)+\omega_{1}E\left(p_{i}\right)+\frac{\omega_{3}}{C(p_{i})}; \end{array} \end{array}$$
where $HT(p_{i})$ represents the minimum link holding time of Equation (Equation 1) on path $p_{i}$, $E(p_{i})$ represents the minimum residual energy of Equation (Equation 4) on path $p_{i}$, and $C(p_{i})$ represents the maximum congestion degree of Equation (Equation 5) on path $p_{i}$. Here, $\omega_{1}$, $\omega_{2}$, and $\omega_{3}$ are the weights, and each weight has a range of values between [0, 1].

### 3.4. Route Process

As shown in Figure 1, the AO-AOMDV protocol achieves route discovery and route maintenance by propagating RREQ messages from the source node S to the destination node D and returning RREP messages from intermediate nodes or the destination node through pre-established reverse paths. Multiple disjoint paths are established between the source and destination nodes, resulting in the discovery of N non-overlapping paths. Assume that we first calculate the fitness value of path p1 (e.g., L1, L2, L3), where $HT(p_{1})$, $E(p_{1})$, and $C(p_{1})$ are the maximum or minimum values among L1, L2, and L3, respectively. This yields $F(p_{1})$. Similarly, we compute the fitness values for other paths, and finally, using AO, we select the highest fitness value among all valid paths. Let us now understand the process of route discovery and route maintenance.

#### 3.4.1. Route Discovery Process

The route discovery process involves the widespread dissemination of RREQ packets, followed by waiting for the route reply (RREP) packets to be unicast from the destination node back to the source node with a timeout determining the completion of the route discovery process. When a node needs to transmit data to another node, it immediately checks the entries in its routing table. If there is a valid path to the destination node, it forwards the data packet to the next hop. Otherwise, it initiates the route discovery process again. The RREQ packet includes fields such as RREQ ID, Originator IP Address, Originator Sequence Number, Destination IP Address, Destination Sequence Number, and Hop Count. The combination of the source address and sequence number (SN) allows for the unique identification of RREQ packets.

When a node receives an RREQ packet, if the routing table does not have an entry for the originator IP address, the current node establishes a reverse path using the sequence number (SN) of the RREQ packet and stores the reverse path entry in its routing table. If the routing table already has an entry for the originator IP address, the SN in the RREQ packet is compared with the SN in the routing table. If the RREQ SN is greater than the SN in the routing table, the existing reverse path entry in the routing table is updated. The RREQ packet is then forwarded by broadcasting it to nearby nodes. If the RREQ SN is not greater, the RREQ packet is discarded. The reverse path entry in the routing table includes the IP address of the receiving node, the source IP address, the hop count to the source node, the source sequence number, and the RREQ source IP address. If the current node is the destination node or an intermediate node with a fresh enough entry, the reverse path is utilized to deliver the RREP to the node that previously received the RREQ. The SN is crucial in avoiding routing loops and determining the freshest entry in the routing table.

Assume that there is an arbitrary node n and a destination node D in Figure 1. When it is necessary to update the routing table, specifically updating the sequence number of D at position i, the corresponding broadcast hop count is initially set. For the sequence number of D, when any node n initializes the routing broadcast for D, the hop count is updated according to Equation (Equation 7), where A represents the broadcast hop count, $hop_{c}ount^{D}nk$ denotes the hop count of the k-th route in the routing table from n to D, that is $\left(last\_count_{nk}^{D},next\_count_{nk}^{D},hop\_count_{nk}^{D}\right)\in route\_list_{nk}^{D}$.
(7)$$\begin{array}{c} \begin{array}{c} A_{n}^{D}=\left\{\begin{array}{cc} max(hop\_count_{nk}^{D}), & n\neq D\\ 0, & otherwise. \end{array}\right. \end{array} \end{array}$$

The routing table of nodes will be updated through received routing control packets, as the traditional AOMDV only selects the optimal route based on the minimum hop count without considering node energy or route congestion. Therefore, we employ a fitness function to assess the routing situation and use arithmetic optimization to select the best route, as described in the routing maintenance process.

#### 3.4.2. Route Maintenance Process

There are two methods for routing maintenance. The first method involves a node broadcasting HELLO messages to neighboring nodes within a one-hop distance. If a node does not receive any messages from a specific neighbor within the configured time interval, it considers the link between the current node and that neighbor node to be disconnected. The second method is responding with a route error (RERR) message. When a node’s neighbor becomes disconnected or a data packet cannot be transmitted to the destination node, the node sends an RERR to its predecessor node. Upon receiving the RERR, the predecessor node marks the route to this destination node as invalid in the routing table and resets its lifetime. If the lifetime expires, the route to this destination node is removed.

In our mechanism, once multiple routes using the AOMDV are received at the source node, we determine the minimum link hold time using Equation (Equation 1), obtain the minimum remaining energy based on Equation (Equation 4), and calculate the maximum congestion degree using Equation (Equation 5). Next, we calculate the fitness value for each route according to Equation (Equation 6) and employ an arithmetic optimization (AO) to select the route with the highest fitness value as the most efficient path.

The AO is a meta-heuristic optimization algorithm that achieves global optimization based on the distribution characteristics of arithmetic operators. Multiplication and division operations improve the global dispersion of position updates, while addition and subtraction operations improve the accuracy of position updates in local regions, as shown in Algorithm 1. $Best_{FF}$ represents the maximum fitness value for each available route between the source node and the destination node.    
**Algorithm 1:** AO Pseudo code
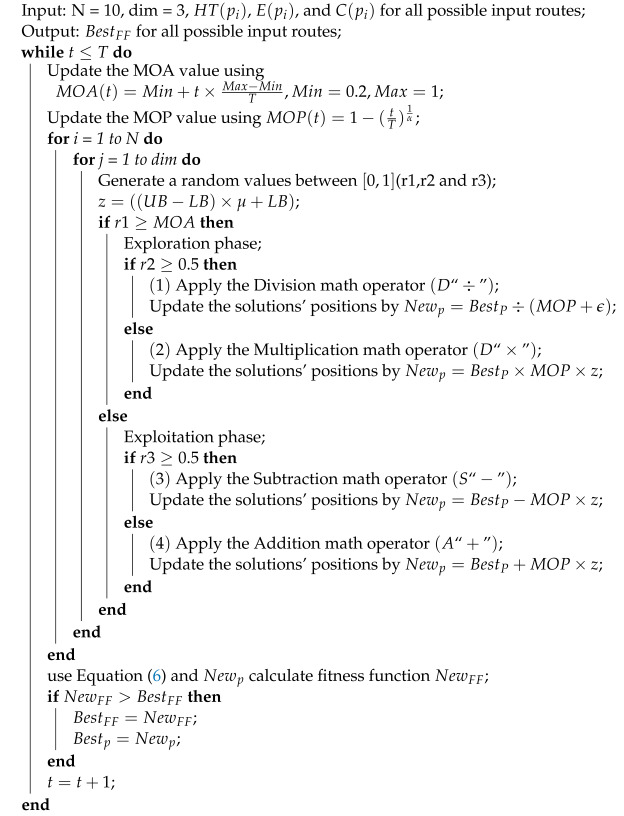


We can simply summarize the AO into three steps as follows.

Step 1, Accelerating function selection optimization strategy through mathematical optimizer.

Step 2, Exploration stage, which uses the multiplication strategy and division strategy for a global search to increase solution diversity, enhance the algorithm’s global optimization ability, and overcome premature convergence, achieving global exploration and optimization.

Step 3, Development stage, which uses the addition strategy and subtraction strategy to reduce solution diversity, which is beneficial for the population to fully exploit the local range and strengthen the algorithm’s local optimization ability.

For a better understanding of the proposed AO-AOMDV protocol, we summarize the step-by-step process of the AO-AOMDV protocol in the following.

(1)Route Discovery Initiation: The source node S initiates the route discovery process by broadcasting an RREQ message to its neighboring nodes.(2)RREQ Propagation: The RREQ message is propagated from one node to another based on the routing protocol’s algorithm. The RREQ message contains information about the source node S, destination node D, and other necessary parameters.(3)Reverse Path Setup: Intermediate nodes and/or the destination node D, upon receiving the RREQ message, establish reverse paths to the source node S. These reverse paths will be used later for returning the RREP message.(4)RREP Generation and Propagation: Once the RREQ message reaches the destination node D, it generates an RREP message. The RREP message is then propagated back to the source node S through the pre-established reverse paths.(5)Multiple Forward Paths Establishment: The source node S and the destination node D establish multiple forward paths between them. These forward paths are designed to be disjoint, ensuring redundancy and reliability.(6)Fitness Function Evaluation: For each of the established forward paths, a fitness function is applied to calculate their fitness values. The fitness function considers factors such as path holding time, energy consumption, congestion degree, etc.(7)AO Path Selection: We will sort the fitness values of all routes calculated through arithmetic optimization in descending order. The route from the source node to the destination node will automatically select the top-ranked route to transmit data packets. In the event of a link disruption, the second-ranked route in the sorting will be chosen to transmit data packets, and so on, in a sequential manner.

## 4. Performance Evaluation

### 4.1. Performance Metrics

We utilize three performance metrics—the packet delivery rate (PDR), average end-to-end delay (E2E), and network lifetime—to demonstrate the effectiveness of the proposed routing protocol. The evaluation is conducted considering variations in the UAV node density and velocity.

(1)PDR: The number of data packets successfully received by the destination node divided by the number of data packets sent by the source node, excluding control packet traffic.(2)Average E2E: The total simulation time divided by the total number of data packets sent by the source node.(3)Network Lifetime: The time taken for the simulation until the first node’s death, with a longer network lifetime indicating more robust routing.

### 4.2. Simulation Environment

In the NS3 simulation, we conducted a comparative analysis between the AO-AOMDV and the classical reactive routing protocols AOMDV and AODV. We employed a Markov Gaussian mobility model for the UAV nodes, which were uniformly distributed in a 3D space of 2000 m × 2000 m × 300 m. The variation in the Z-axis ranged from 100 m to 300 m. The UAV speeds ranged from 10 m/s to 40 m/s. The maximum transmission range for the UAVs was set to 250 m. At the MAC layer, we utilized the IEEE 802.11n protocol. At the application layer, a random UAV node was selected as the server, while other UAV nodes served as clients, taking turns sending data packets to the server. The detailed parameters employed in our simulation are summarized in Table 1.

### 4.3. Simulation Results and Discussion

In this section, the simulation results of the AO-AOMDV algorithm were analyzed and compared. Figure 2 illustrates the spatial distribution of the UAV nodes, with 100 UAV nodes uniformly distributed within the specified area. In our simulation, the UAV nodes dynamically and randomly vary within a fixed range of 3D space at a certain rate of change.

#### 4.3.1. Impact of UAV Node Velocity

We investigated the impact of different node velocities in FANETs. The velocities of the UAV nodes varied between 10 m/s and 40 m/s, while the number of UAV nodes remained fixed at 100.

Figure 3 illustrates the PDR of the compared routing protocols for varying UAV velocities. When the movement velocity is less than 15 m/s, the probability of link breakage is low, as the packet delivery ratios of the three protocols are similar and maintained above 95%. As the node velocity increases, the packet delivery ratios of the three protocols continuously decrease. In comparison, the AO-AOMDV calculates the optimal route using an arithmetic optimization algorithm, which effectively reduces the number of control packets for route rediscovery and decreases the node buffer occupancy. In contrast, other traditional routing protocols use the shortest route, which can lead to congestion and packet loss, resulting in performance degradation. Therefore, the AO-AOMDV demonstrates a significantly higher packet delivery ratio compared to traditional algorithms. When the velocity exceeds 35 m/s, the probability of link breakage increases due to excessive speed, leading to a faster decline in the packet delivery ratios of the three protocols.

Figure 4 indicates the average E2E delay of the compared routing protocols for varying UAV velocities. When the movement velocity is less than 20 m/s, the delay of the AO-AOMDV protocol is slightly higher than that of the traditional protocols. This is due to the presence of the arithmetic optimization algorithm, which leads to a slightly higher number of hops in the AO-AOMDV protocol, resulting in additional forwarding delay. As the velocity increases, nodes frequently enter and exit, and once a link breakage occurs along the path, the route discovery process needs to be repeated, increasing the discovery delay, which becomes more pronounced with higher velocities. The AO-AOMDV utilizes the arithmetic optimization algorithm to reduce the probability of link breakage and avoid path congestion. Therefore, when the velocity exceeds 30 m/s, the AO-AOMDV demonstrates good performance in terms of delay.

Figure 5 shows the network lifetime of the compared routing protocols for varying UAV velocities. With the increase in node movement velocity, rapid mobility benefits the average energy consumption of nodes, leading to a gradual increase in the lifetime of the three protocols. Due to the consideration of remaining node energy in the AO-AOMDV, the arithmetic optimization algorithm protects nodes with lower energy during path selection in the later stages of network transmission, thereby maximizing network lifetime.

#### 4.3.2. Impact of UAV Node Density

We also investigated the impact of different node densities in FANETs. The number of UAV nodes varied between 20 and 100, while the velocity of the UAV nodes remained fixed at 10 m/s.

Figure 6 presents the PDR of the compared routing protocols for different amounts of UAV nodes. Initially, due to the low density of the UAV nodes, the network experienced frequent disconnections, resulting in a low packet delivery ratio. However, as the number of UAV nodes increased, the network connectivity improved, leading to an increase in the packet delivery rate. Throughout this process, the AO-AOMDV consistently outperformed traditional routing protocols in terms of the packet delivery rate.

Figure 7 shows the average E2E delay of the compared routing protocols for different amounts of UAV nodes. In low-density networks, it is difficult to form a sufficient number of paths to utilize the arithmetic optimization algorithm, resulting in the AO-AOMDV exhibiting the highest latency. However, as the number of UAVs increases, the latency of all three protocols decreases and tends to stabilize.

Figure 8 indicates the network lifetime as the number of nodes increases. When the number of nodes exceeds 70, the lifetime of the AO-AOMDV protocol surpasses that of the traditional protocols and tends to stabilize.

## 5. Conclusions

In this study, an arithmetic optimization was used to optimize the route discovery process of the AOMDV protocol. A fitness function was constructed by integrating four metrics: the link holding time, residual energy, congestion degree, and hop count. The optimal fitness function was employed for path selection. The simulation results demonstrate that the AO-AOMDV protocol not only adapts well to the high mobility of the UAV nodes and drastic changes in topology but also provides better service quality and significantly prolongs the lifetime of UAV networks. Although the algorithms involved in this research introduce certain parameters as routing selection criteria, the covered metrics are not comprehensive. As part of our future work, we will consider incorporating indicators such as available bandwidth and jitter to design routing protocols.

## Figures and Tables

**Figure 1 sensors-23-07550-f001:**
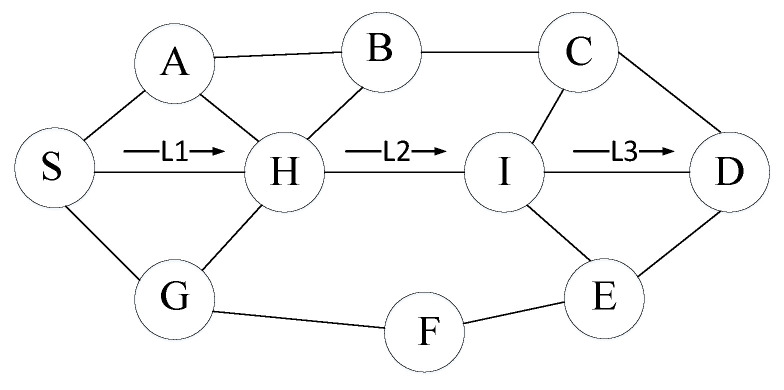
AOMDV routing protocol.

**Figure 2 sensors-23-07550-f002:**
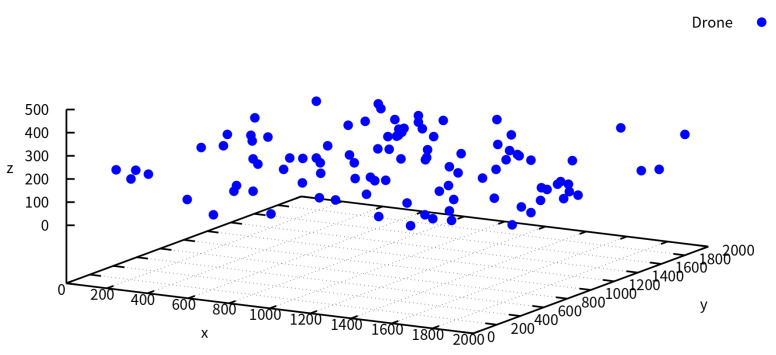
3D deployment of UAV nodes for simulation.

**Figure 3 sensors-23-07550-f003:**
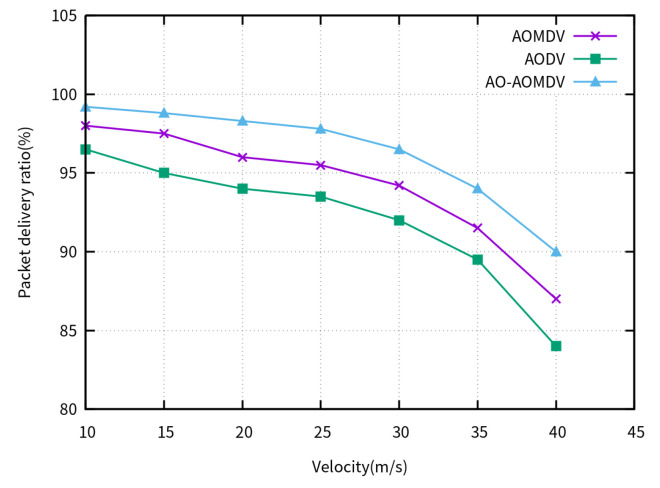
PDR for varying UAV velocities.

**Figure 4 sensors-23-07550-f004:**
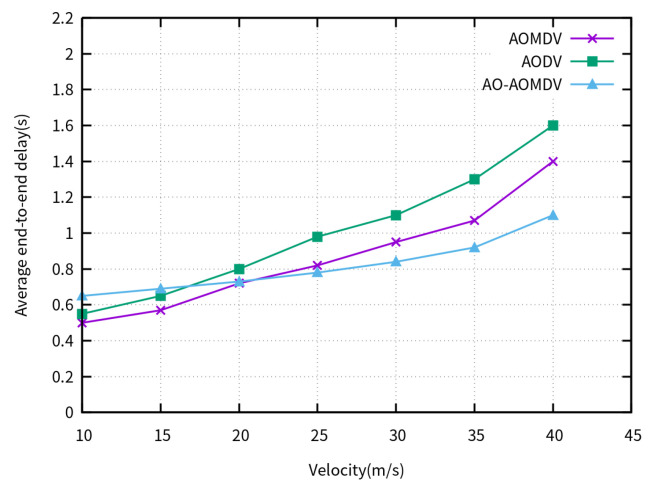
Average end-to-end delay for varying UAV velocities.

**Figure 5 sensors-23-07550-f005:**
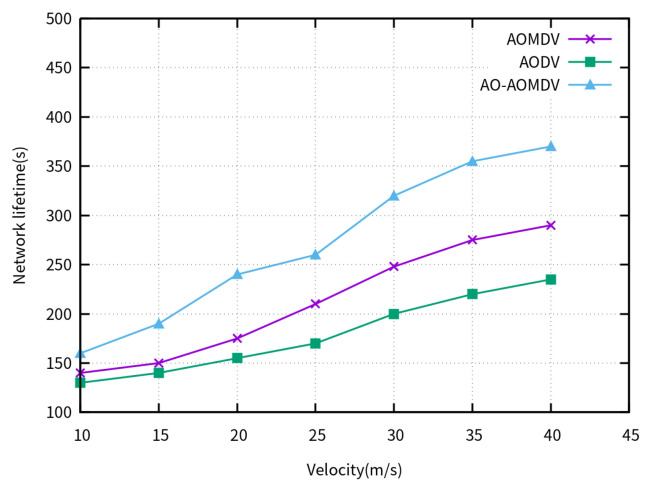
Network lifetime for varying UAV velocities.

**Figure 6 sensors-23-07550-f006:**
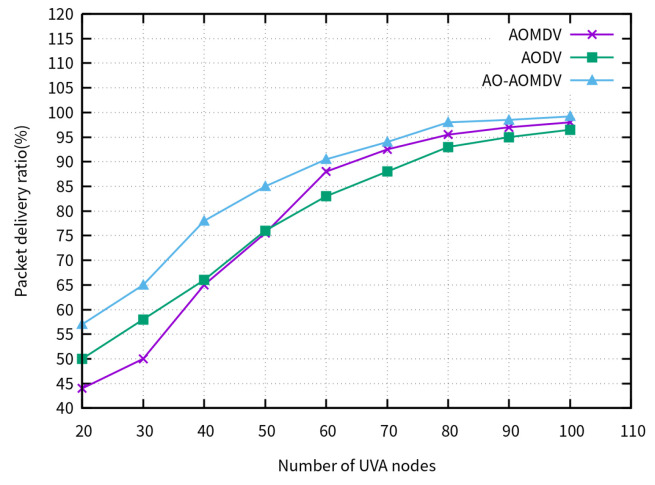
PDR for varying node densities.

**Figure 7 sensors-23-07550-f007:**
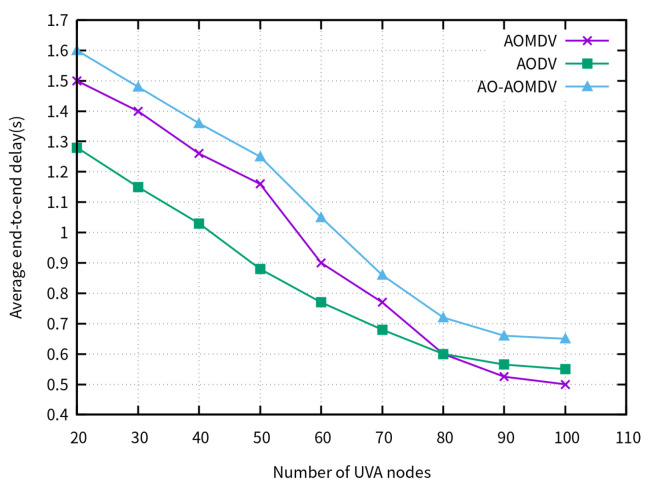
Average E2E delay for varying node densities.

**Figure 8 sensors-23-07550-f008:**
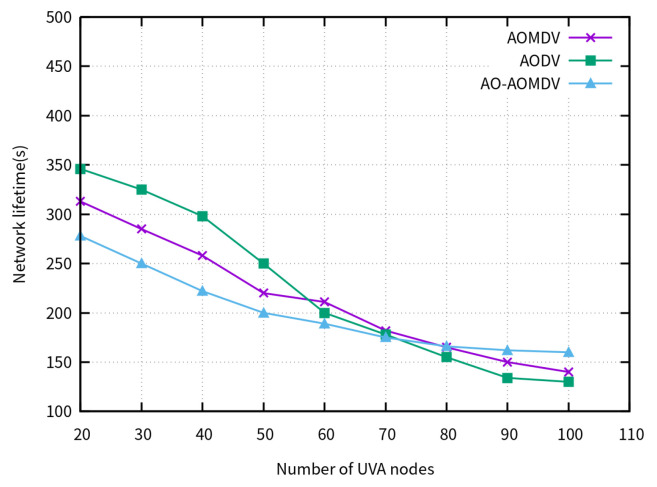
Network lifetime for varying node densities.

**Table 1 sensors-23-07550-t001:** Simulation parameters.

Parameter	Value
Simulator	NS3
Maximum number of nodes	100
3D network dimension	2000 m × 2000 m × 300 m
MAC protocol	IEEE 802.11n
Bandwidth	20 MHz
Velocity	10–40 m/s
CBR rate	1 Mbps
Transport protocol	UDP
Mobility model	3D Gauss Markov mobility model
Simulation time	600 s
Compared routing protocol	AOMDV, AODV

## Data Availability

Not applicable.

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
