# Peer review of "Arithmetic Optimization AOMDV Routing Protocol for FANETs"

_sensors, 2023, doi:10.3390/s23177550_

Round 1

Reviewer 1 Report

This manuscript presented an arithmetic optimization method for flying ad hoc networks. There are several points that need further consideration and special attention.

  1. The authors considered the energy consumption of communication. How does this consumption compare to the energy consumption of the UAVs? In my opinion, the energy consumption of communication modules is usually not significant compared to the consumption of the UAVs if the UAVs are electric. 
  2. The first concern directly leads to the second question: what type of UAVs was considered? How were they powered? Are they fixed-wing or multi-rotor?
  3. Why is the energy modeled by Eq. (2)? Are there any references for it?
  4. If the UAVs are moving at different speeds or directions, their distance will change all the time. Will the distance cause disconnection and time delays? Should these factors be considered?

Please proofread carefully.

Author Response

Dear Review:

I am very pleased to receive your editorial comments. In response to your queries, I provide the following explanations.

1.The authors considered the energy consumption of communication. How does this consumption compare to the energy consumption of the UAVs? In my opinion, the energy consumption of communication modules is usually not significant compared to the consumption of the UAVs if the UAVs are electric.

Solution:  I am grateful for the suggestions you provided, but I agree with your viewpoint concerning individual drones. However, for tasks that require collaboration among multiple drones, based on simulation results, the routing protocol proposed in this paper can still maximize the potential extension of the drone network's lifespan. Specifically, this protocol plays a proactive role in balancing energy consumption among drones, thereby facilitating the deployment of a rotating operational scheme within the drone network. Thank you.

2.The first concern directly leads to the second question: what type of UAVs was considered? How were they powered? Are they fixed-wing or multi-rotor?

Solution:  I appreciate your inquiry. However, it's worth noting that the routing protocol proposed in this paper is not strongly correlated with the type of drone. Its relevance primarily emerges when drones require self-organizing networks for wireless communication. If power consumption is taken into consideration, the response aligns with the answer provided in response to Question 1. Thank you.

3.Why is the energy modeled by Eq. (2)? Are there any references for it?

Solution: As commercial drones currently utilize WiFi for wireless communication, I have referenced the energy calculations from the switching of operational modes in the 802.11n standard for WiFi wireless networks.

4.If the UAVs are moving at different speeds or directions, their distance will change all the time. Will the distance cause disconnection and time delays? Should these factors be considered?

Solution:  I have explained the parameter selection in subsection 3.2 as per your suggestions.

All explanations are as stated above. I hope you can find time to review them. If you have any further suggestions, please feel free to let me know at your earliest convenience.

Sincerely yours,
Huamin Wang
[email protected]

Reviewer 2 Report

In this research, the authors proposed an optimization method for a routing protocol between flying drones. This method uses a fitness function to calculate the fitness values of multiple paths and employs an arithmetic optimization to select the optimal route for routing selection. The effectiveness of the proposed method is demonstrated by simulation results. It achieves the high packet delivery ratio, lower average end-to-end delay and network lifetime. 

In the current version, the authors presented the related works for the routing parameters and optimization methods commonly used by researchers. However, they did not explain their proposed method in details and its advantages over other methods.  Below are my questions and comments on this paper:

1. In the introduction, the authors described the background of the research in detail. However, they have not properly explained the proposed method and the contributions of this paper.  The authors should emphasize them. In addition, it would be better if the authors add a new paragraph to explain the main content of each section.

2. All abbreviations should be explained in the first section when mentioned. For example, the terms FANET and MANET should be explained in the introduction section.

3. The author proposed the optimization method for the routing protocol on flying drones. However, the authors did not clearly explain each parameter, why the authors chose them over others, and how these parameters will affect the performance.

4. Figure 2 and its explanation were not related.

5. The authors should add the input and output variables in Algorithm 1. This will help the readers to easily understand the proposed optimization method.

6. In conclusion, the authors mentioned "QoS parameters as routing selection criteria". However, there is no explanation of "QoS parameters" in the sections of the current paper. The authors should provide a description of these parameters.

7. The English language used in the article needs to be polished.

8. The authors should consider the standard format for the table.

9. A large blank space appear in subsection 3.4.2. It would be better if the author removed this gap.

10. The performance of the proposed method was evaluated only by simulations. However, the implementations using real drones in real fields are often different from simulations. It is necessary and very important to show how the real implementations are different from simulations and how the differences can be overcome or ignored. Simulations only are not sufficient as an technical paper for engineering

The English language used in the article needs to be polished

Author Response

Dear Review:

I am very pleased to receive your editorial comments. In response to your queries, I provide the following explanations.

  1. In the introduction, the authors described the background of the research in detail. However, they have not properly explained the proposed method and the contributions of this paper. The authors should emphasize them. In addition, it would be better if the authors add a new paragraph to explain the main content of each section.

Solution:  I have incorporated emphasis and explanation into the final two paragraphs of the introduction, as per your suggestions.

  1. All abbreviations should be explained in the first section when mentioned. For example, the terms FANET and MANET should be explained in the introduction section.

Solution:  I have provided explanations for the terms in the introduction as per your suggestions.

  1. The author proposed the optimization method for the routing protocol on flying drones. However, the authors did not clearly explain each parameter, why the authors chose them over others, and how these parameters will affect the performance.

Solution:  I have explained the parameter selection in subsection 3.2 as per your suggestions.

  1. Figure 2 and its explanation were not related.

Solution:  I have removed Figure 2 as per your suggestions.

  1. The authors should add the input and output variables in Algorithm 1. This will help the readers to easily understand the proposed optimization method.

Solution:  I have added input and output variables to Algorithm 1 as per your suggestions.

  1. In conclusion, the authors mentioned "QoS parameters as routing selection criteria". However, there is no explanation of "QoS parameters" in the sections of the current paper. The authors should provide a description of these parameters.

Solution:  I have modified the conclusion because my wording was incorrect.

  1. The English language used in the article needs to be polished.

Solution:  I intend to seek the assistance of a professional editor for proofreading.

  1. The authors should consider the standard format for the table.

Solution:  Yes, I have resolved it along with the other issues.

  1. A large blank space appear in subsection 3.4.2. It would be better if the author removed this gap.

Solution:  Yes, I have resolved it along with issue 4.

  1. The performance of the proposed method was evaluated only by simulations. However, the implementations using real drones in real fields are often different from simulations. It is necessary and very important to show how the real implementations are different from simulations and how the differences can be overcome or ignored. Simulations only are not sufficient as an technical paper for engineering

Solution:  I fully agree with your perspective. I am an engineer who places a strong emphasis on practical implementation, and over the past two years, I have spent most of my time coding in areas such as Xilinx (FPGAs), QT interfaces, and VB.NET. The engineering implementation of drones is a massive undertaking that heavily relies on team collaboration and financial support. In the future, if I have the opportunity to delve further into research within the drone field, I will definitely invest in complementary equipment to apply my findings. For now, I can only strive to adjust simulation parameters to align them as closely as possible with the real world.

All explanations are as stated above. I hope you can find time to review them. If you have any further suggestions, please feel free to let me know at your earliest convenience.

Sincerely yours,
Huamin Wang
[email protected]

Round 2

Reviewer 1 Report

The authors have addressed all the issues. I have no further questions.

Author Response

Dear Review:

       Thank you very much for your recognition. Of course, our paper still has areas that need refinement and improvement. I have already made further revisions based on the feedback from another reviewer. If you have any additional suggestions, please feel free to share them with me at your convenience.

Sincerely yours,
Huamin Wang
[email protected]

Reviewer 2 Report

According to the revised paper and the response letter, the current version is an improvement over the previous version. However, the authors did not properly address my suggestions, and their response letter did not clearly answer some of my questions. Below are my questions and comments on this revised version: 

1. I appreciate that the authors considered my suggestions. The authors have presented the parameters to be optimized by the proposed method.  However, it would be better if the authors presented them in complex paragraphs. The current version is insufficient for a scientific journal.

2. Authors should consider the standard MDPI format for tables. I found there is no modification compared to the previous version.

3. Authors must improve the presentation quality of this manuscript.  I have found typographical errors and inconsistencies in the use of a list style to explain certain processes. The manuscript should be checked carefully before submission.

4. Regarding my previous question about the real implementations to evaluate the proposed method, the author's response letter did not clearly address this question.

The English language used in the article needs to be polished

Author Response

Dear Review:

I'm very glad to receive your suggestion again, and I sincerely apologize for any omissions and unclear parts in the revisions.

  1. I appreciate that the authors considered my suggestions. The authors have presented the parameters to be optimized by the proposed method.  However, it would be better if the authors presented them in complex paragraphs. The current version is insufficient for a scientific journal.

Solution: Thank you very much for your suggestions. However, the routing protocol I proposed is primarily built upon classical routing protocols and is optimized for highly mobile drone networks. Similar to question 4, it represents a combination of science and engineering, taking into account practical performance and cost considerations. I do intend to continuously enhance my theoretical knowledge and engineering expertise in the future, in order to apply my research paper to real-world projects.

  1. Authors should consider the standard MDPI format for tables. I found there is no modification compared to the previous version.

Solution: I apologize for previously omitting Table 1. I have now updated Table 1 according to MDPI's standard template.

  1. Authors must improve the presentation quality of this manuscript.  I have found typographical errors and inconsistencies in the use of a list style to explain certain processes. The manuscript should be checked carefully before submission.

Solution: Following your advice, I have made adjustments and revisions to all list styles. If there are any remaining concerns, I would greatly appreciate your detailed feedback. Thank you.

  1. Regarding my previous question about the real implementations to evaluate the proposed method, the author's response letter did not clearly address this question.

Solution: Thank you very much for your input. However, there are the following three reasons: Firstly, articles related to drone ad hoc network routing protocols mostly involve simulation studies, and I haven't come across any examples of real-world implementations that could serve as references. Next, I have chosen to use a three-dimensional mobility model that better approximates real-world scenarios ,while many papers employ two-dimensional mobility models. Additionally,in the early stages, our funding requirements mandated the publication of a paper to establish the theoretical foundation. In the later phases, we plan to implement it in a real drone network.Thank you.

All explanations are as stated above. I hope you can find time to review them. If you have any further suggestions, please feel free to let me know at your earliest convenience.

Sincerely yours,
Huamin Wang
[email protected]